# The Effects of Egg- and Substrate-Associated Microbiota on the Larval Performance of the Housefly, *Musca domestica*

**DOI:** 10.3390/insects15100764

**Published:** 2024-10-01

**Authors:** Rasmus Majland Dyrholm, Pernille Arent Simonsen, Cino Pertoldi, Toke Munk Schou, Asmus Toftkær Muurmann, Simon Bahrndorff

**Affiliations:** 1Department of Chemistry and Bioscience, Aalborg University, Fredrik Bajers Vej 7H, 9220 Aalborg, Denmark; rdyrho21@student.aau.dk (R.M.D.); pasi21@student.aau.dk (P.A.S.); asmustm@bio.aau.dk (A.T.M.); sba@bio.aau.dk (S.B.); 2Aalborg Zoo, Mølleparkvej 63, 9000 Aalborg, Denmark; 3ENORM Biofactory, Hedelundvej 15, 8762 Flemming, Denmark; tms@enormbiofactory.com

**Keywords:** insect production, microorganisms, egg disinfection, substrate autoclaving, survival rate, biomass, growth

## Abstract

**Simple Summary:**

The increase in human population size has resulted in an increase in the demand for food for human consumption and feed for livestock. Insects have been suggested to be a more sustainable food and feed source and are, thus, receiving growing interest. Insects, such as the housefly (*Musca domestica* Linnaeus), can be used to valorise different waste and by-products from the agricultural industry and are also more effective in converting feed into biomass than conventional livestock. This study aims to identify how microbial communities affect growth, survival rate, and biomass of the house fly. This is examined by disinfecting fly eggs, autoclaving the growth substrate, or both. Results show that a reduction in microorganisms from eggs and substrate results in reduced growth and biomass. However, disinfecting eggs does not result in a lower survival rate, but autoclaving the substrate reduces the survival rate. This study highlights the importance of microbial communities’ presence and how they influence house fly’s performance.

**Abstract:**

Increasing human population size and income growth are causing an increasing demand for food and feed. Insects are a more sustainable alternative to conventional animal source proteins, as they can convert waste and by-products from the agricultural industry into biomass for commercial feed for livestock and, potentially, serve as a food source for human consumption. Moreover, insects together with their microorganisms have been shown to play a pivotal role in the development of insects and in the breakdown of complex growth substrates, and are, therefore, closely tied to insect production. This study aims to determine if the removal of egg- and substrate-associated microorganisms impacts larval performance (growth, final biomass, and the survival rate) of *M. domestica* Linnaeus. Four treatments are tested: disinfected eggs and non-autoclaved substrate, non-disinfected eggs and autoclaved substrate, disinfected eggs and autoclaved substrate, and a control without any removal of microbiota. The results show a significant decrease in the final biomass of larvae subjected to the treatments with only disinfected eggs, only autoclaved substrate, and both compared to the control, and a significant decrease in survival rate for non-disinfected eggs and autoclaved substrate and disinfected eggs and autoclaved substrate compared to the control group. Moreover, larval growth shows a significant difference across days within all treatments. Together, this suggests that the microorganisms of housefly eggs and the growth substrate play an important role in biomass, which is critical in commercial insect production. Together this suggest, that more studies are needed to examine these parameters with respect to more commercially relevant substrates.

## 1. Introduction

Rapid global population growth exerts pressure on agricultural production for food and feed [1]. It is estimated that, by 2050, food production will need to increase by 70% compared to 2009, and the demand for proteins of animal origin will have increased by 74% [2]. The agricultural industry is currently using 37% of Earth’s land mass for food production. This causes deforestation and, as a result, it inflicts a loss of biodiversity and increased greenhouse gas emissions [3]. Alone, livestock production is responsible for 14% (excluding land use) of the world’s greenhouse gas emissions. These emissions originate from manure management, ruminant waste on pastures, ruminant emissions, and fertiliser production [3]. Moreover, food loss and waste account for 8% of the global greenhouse gas emissions [3]. In addition, results suggest that the agricultural industry is responsible for 92% of global water consumption [3]. These problems necessitate proper handling of waste- and by-products and a sustainable quality protein source for food and feed production.

Insects are an interesting alternative solution for the abovementioned agricultural-related challenges, as they enable the conversion of waste and by-products from the agricultural industry into biomass that can potentially be used for commercial feed for livestock animals, as a potential food source for humans [4,5], and as fertiliser producers [6]. Insects both have a high feed conversion rate in comparison to livestock [7,8] and can be used to valorise waste and by-products, such as used grains, household waste, and manure [9]. For example, manure substrates have been found to be effective feed for the housefly (*Musca Domestica* Linnaeus) and the black soldier fly (*Hermetia illucens* Linnaeus), and other insect species have been postulated to be effective in utilising manure as feed as well [5].

It has been suggested that host-associated microorganisms, together with the host, play an important role in the breakdown of low degradable substrates [9], but the role of each is currently unclear. Microorganisms play a critical role in the fitness of insects [10], where some bacterial species exist as either endosymbionts or ectosymbionts, and can exist in a mutualistic, commensalistic, or pathogenic association with the host [11]. Microorganisms can be acquired in various ways, such as vertically from mother to offspring [12], or horizontally transmitted through substrates [13,14]. Furthermore, free-living bacteria can be acquired throughout the life cycle. Vertically transmitted microbes have been shown to be conducive to increased fitness through influence on factors like reproductive ability or metabolic performance [10,15].

Host-associated bacteria are predominantly present in the digestive system, where they play a role in key aspects of their host’s metabolism, primarily aiding in the acquisition of nutrients such as nitrogen, vitamins, and sterols [11,16]. For example, in honeybees (*Apis mellifera* Linnaeus), the bacteria *Gilliamella apicola* helps the breakdown of pollen. Additionally, the bacteria *Ishikawaella capsulata* aids the growth and reproduction of the Japanese common plataspid stinkbug (*Megacopta punctatissima* Montandon) and is also found to contain pathways for many essential and non-essential amino acids [17]. Bacteria have been shown to benefit the development of *M. domestica*, as they aid in nutritional acquisition and metabolism of nutrients in organic substrates or can serve as a nutrient source [18]. Larvae of *H. illucens* have also shown dependence on microorganisms in the environment, since substrate-associated microorganisms were found to influence the survival, weight, and proportion of pre-pupae in *H. illucens* [19]. Furthermore, microorganisms in insects can improve resistance to environmental disturbances, enhance longevity, and shorten larval development time [20]. Further, for *M. domestica,* it has been determined that different bacteria are present at different life stages. For example, the bacteria strain *Weissella* dominates the colonisation of newly hatched larvae, while, the second larval stage is dominated by strains of *Stenotrophomonas*, *Bacillus*, and *Lactococcus*. This suggests that the bacteria have different impacts during different life stages [21]. A specific example of this is the bacteria *Klebsiella oxytoca*’s ability to restrict further oviposition at sites with recently laid eggs for female flies [21].

In addition to their presence in larval intestines, bacteria are also present on the surface of eggs. Mazza et al. (2020) isolated nine different bacteria present on the eggs of *H. illucens* and specific combinations of the bacteria could either negatively or positively impact the weight gain and conversion rates in their larval stage [22]. Eggs are also sensitive to pathogens and the host can, via microorganisms, inhibit pathogens [10]. It is, therefore, not surprising that several papers have suggested that bacteria play a key role in commercial insect production [9,23].

*M. domestica* is an organism which can be used to determine the effects of vertically transmitted and substrate-associated microorganisms. In addition, their larvae are promising candidates as an alternative food- and feed-source [24]. This study aims to quantify the influence of the egg- and substrate-associated microorganisms on *M. domestica* larvae’s performance.

## 2. Materials and Methods

The effect of microorganisms on larval performance will be investigated through autoclaved substrates (a combination of wheat bran, *Triticum aestivum*, alfalfa, *Medicago sativa*, and water) and surface-disinfected eggs using a sodium hypochlorite solution. Tests were performed to examine the effects on the performance of *M. domestica* larvae through the following parameters: growth, final biomass, and survival rate.

### 2.1. Study Organism and Rearing

The flies used in the present study were obtained from a laboratory culture at Aalborg University, established from flies collected from Danish farms in 2019 and since kept in the laboratory. The culture was kept at 23 ± 1 °C under a cycle of 12 h of light and 12 h of darkness. The larvae were reared on a substrate consisting of 65.75% water, 21.92% wheat bran (*Triticum aestivum* Linnaeus), 10.96% alfalfa (*Medicago sativa* Linnaeus), 0.82% maltose, and 0.55% dry yeast, with a density of 0.67 eggs per gram substrate. When pupae were observed, they were placed in a rearing cage (BugDorm-1 Insect Rearing Cage, MegaView Science Co, Taichung City, Taiwan). When adult flies emerged, water was available ad libitum and the flies were fed a mixture consisting of sugar, powdered sugar, and milk powder, all of which were available ad libitum throughout the adult stage.

### 2.2. Experimental Design

To test how substrate-level microorganisms and the maternally inherited egg-surface-level microbes affect the growth, final biomass, and survival rate, two experiments were conducted.

Preliminary test: the preliminary test was conducted to determine which concentration of chlorine bleach, and, more specifically, which concentration of the active substance 2.5% *w/w* sodium hypochlorite, was optimal to ensure the survival of the eggs until the larval stage while also reducing microbiota. Six treatments of various sodium hypochlorite concentrations were used: 0% *w*/*w*, 0.5% *w*/*w*, 0.625% *w*/*w*, 0.825% *w*/*w*, 1.25% *w*/*w*, and 2.5% *w*/*w*, see Appendix A for calculations (25 mL water and 0 mL bleach, 100 mL water and 25 mL bleach, 75 mL water and 25 mL bleach, 50 mL water and 25 bleach, 25 mL water and 25 mL bleach, 0 mL water and 25 mL bleach, respectively). For each of the six treatments, 50 eggs were collected from the laboratory culture and were disinfected.

Both in the preliminary test and the main experiment, the egg surface was disinfected with chlorine bleach (0.5% sodium hypochlorite in the main experiment). The disinfection process was conducted by placing the eggs in a sieve and by disinfecting the surface by washing the eggs with chlorine bleach and letting it sit for 4 min. Thereafter, the chlorine bleach was rinsed with ultra-pure water and the eggs were transferred to a container using ultra-pure water. A disinfected fabric was used to cover the containers of the disinfected eggs and substrate. The process took place in a laminar flow cabinet. In the containers, there were 50 g of autoclaved substrate (consisting of 10.81% alfalfa, 21.62% wheat bran, and 67.57% water). The substrate was autoclaved in a 1000 millilitre glass container at 120 °C in an autoclave for 2 h.

To determine the effect of disinfection on the hatch rate, larvae were counted after six days.

Further, one egg was placed on a plate of LB agar and incubated aerobically to examine the effectiveness of the disinfection. The LB agar plates were prepared in a 250 mL glass container with, 5 g/L agar (VWR, Leuven, Belgium), 2.5 g/L NaCl (VWR, United States of America), 2.5 g/L tryptone (Sigma Aldrich, Steinheim, Germany), 1 g/L yeast extract (Thermo Scientific, Waltham, MA, USA), and 250 mL of water was added. The pH was adjusted to 7.0 with NaOH. The LB agar medium was then autoclaved and poured into petri dishes. Eggs and their substrate were placed and left on the agar plates under sterile conditions in a laminar flow cabinet. The plates were incubated at 35 °C and examined after five days. The number of colonies was counted by visual inspection or with the use of a grid to count colonies methodically. These were compared with hatching rates to evaluate the effectiveness of each solution. Hatching rates were calculated using the following formula:$$Hatching\;rate\;\left(\%\right)\;=\;\left(\frac{Number\;of\;hatched\;eggs}{Total\;number\;of\;eggs\;laid}\right)\cdot 100\%$$

The number of colonies and the corresponding hatching rates are shown in Appendix A, Table A1 for experiment 1. The number of colonies for experiment 2 is shown in Appendix B, Table A3.

The main experiment: based on the preliminary test, it was found that a concentration of 0.5% *w*/*w* sodium hypochlorite had the highest rate of survival while also reducing the microbiota on the surface of the eggs. In the main experiment, four different treatments were prepared: (A) one with disinfected eggs and non-autoclaved substrate, (B) one with non-disinfected eggs and autoclaved substrate, (C) one with disinfected eggs and autoclaved substrate, and (D) a control with no disinfection and no autoclaving of eggs and substrate. Thus, 80 containers, each representing a sample, were prepared: 40 with 50 g of autoclaved substrate and 40 with 50 g of non-autoclaved substrate. A total of 8000 eggs were collected: 4000 were disinfected with 0.5% *w*/*w* sodium hypochlorite and 4000 were not. Each treatment consisted of 20 replicates for each 100 eggs with 50 g of substrate. The containers were covered with disinfected and non-disinfected fabric depending on the treatment and placed under the same conditions as those used for the laboratory culture. Every day, the containers were stirred.

To examine the presence of microbiota across treatments, four LB agar plates were produced and, subsequently, 25 disinfected eggs, 25 non-disinfected eggs, autoclaved substrate, or non-autoclaved substrate were placed on the plates, see Appendix B, Figure A2.

#### Determination of Growth, Final Biomass, and Survival Rate

In the main experiment, when five larvae or more were observed in a given replicate, larvae were subsequently weighed on a daily basis. This continued until the day when a decrease in the mean larvae biomass under a given treatment was observed. Subsequently, the surviving larvae and pupae were frozen and then counted to determine the rate of survival. The larvae were weighed individually after being frozen to determine the final biomass. For samples where less than five larvae were present, the larvae were only used for the determination of the final biomass and rate of survival. The survival rate was calculated using the following formula:$$Survival\;rate\;\left(\%\right)\;=\;\left(\frac{Number\;of\;larvae\;and\;pupae}{100}\right)\cdot 100\%$$

### 2.3. Statistical Analysis

All figures and analyses were conducted in R version 4.4.0 using RStudio version 2024.04.1+748 (http://www.rstudio.com, accessed on 4 September 2024) and using the following packages: ggplot2 [25], tidyverse [26], dplyr [27], dunn.test [28], and rstatix [29]. Boxplots were made to visualise the growth, final biomass, and survival rate. A Shapiro–Wilk test confirmed that the data followed a non-normal distribution. Therefore, non-parametric tests were used. A Kruskal–Wallis rank sum test was performed to determine if there was a significant difference among treatments, followed by a Dunn’s test with a Bonferroni correction to determine the significant differences among the treatments. Furthermore, a repeated measures ANOVA was used to assess the effect of time on the growth.

## 3. Results

### 3.1. Growth

Disinfected eggs and/or autoclaved substrate resulted in a slower growth (Figure 1). Furthermore, for A (disinfected eggs) and D (control), the median biomass increased from day to day for the first five days. For B (autoclaved substrate), the median biomass increased during the first four days and from day five to day seven. For C (disinfected eggs and autoclaved substrate), the median biomass increased during the first two days and from day four to day six.

There was a significant effect of time on the growth under all treatments (Table 1). The *p*-values of the repeated measures ANOVA showed a significant difference in the means for the days under all treatments. Furthermore, for A and D, a high F-value indicated a large variance of means among groups in comparison to the variance of means within groups. The smaller F-value for B and C indicated less variance among groups compared to the variance of the means within groups. Moreover, the generalised eta squared (GES) value indicated that 76% and 74% of the variance for A and D, respectively, was explained by time, whereas 36% and 7% of the variance for B and C, respectively, was explained by time.

### 3.2. Final Biomass for Individual Larvae

Disinfected eggs and/or autoclaved substrate resulted in a lower final biomass of larvae compared to the control (Figure 2). There was a significant effect of treatment on the final biomass of the larvae (Kruskal–Wallis: *p*-value < 2.2·$10^{-16}$).

A significant difference in means for the final biomass of the larvae was found among all treatments (Table 2).

### 3.3. Final Biomass for Sample Medians

Disinfected eggs and/or autoclaved substrate resulted in a lower final biomass for larvae compared to the control (Figure 3). There was a significant effect of treatment on the final biomass of the larvae (Kruskal–Wallis: *p*-value < 4.481·$10^{-6}$).

A significant difference in medians for the final biomass of the larvae was found between A and C, B and D, and C and D. There was only a significant difference between A and B before adjusting for multiple comparisons (Table 3).

### 3.4. Survival Rate

Treatment C had a lower median survival rate than the other treatments, B had a lower median survival rate than both A and D, and treatment A had the highest median survival rate (Figure 4). At least one treatment had a significant impact on the survival rate of the larvae (Kruskal–Wallis: *p*-value < 1.002·$10^{-11}$).

A significant difference in means for the survival rate of the larvae was found between A and B, A and C, B and D, and C and D. There was no significant difference between B and C and A and D (Table 4).

## 4. Discussion

Microorganisms can affect insect fitness in various ways [16]. The presence of microorganisms on eggs and in the environment can impact different factors like growth, biomass, and survival rate [11,19,20,30]. However, few studies exist on the impact of microbes on the growth of *M. domestica*, a species used in insect production. It was expected that disinfection of eggs, autoclaving of substrate, or both would lead to decreased growth, reduced final biomass, and impaired survival rate for larvae of *M. domestica* compared to non-disinfected and non-autoclaved conditions (control). A significant difference in biomass across days was found within each treatment. For disinfected eggs and/or autoclaved substrate, a reduced final biomass for the larvae was found. Furthermore, the two treatments with only autoclaved substrate and both disinfected eggs and autoclaved substrate led to a decreased survival rate.

### 4.1. The Effect of Disinfected Eggs

It was expected that disinfected eggs would lead to decreased growth, reduced biomass, and impaired survival. This current study found a significant difference between disinfected- and non-disinfected eggs across days in terms of growth and final biomass. However, this was only the case when larvae within treatments were treated as independent samples. When taking the medians of the samples, a significant difference was no longer found compared to the control, although the same pattern was observed. No significant difference was found for survival rate.

Schreven et al. (2021) examined the effect that a sterile egg surface of *H. illucens* has on the performance, which included final biomass and survival rate. Contrary to the current study, they found no significant effect of sterile eggs on larval survival and biomass. They argued that the microbiota present on the eggs was so limited compared to the number of substrate-associated microorganisms that they did not affect the overall microbiota composition nor the larval performance [19]. However, Mazza et al. (2019) examined the specific egg-associated bacteria present on the eggs of *H. illucens*. They found that the bacteria were able to enhance biomass and influence the accumulation of crude fat and crude protein depending on the specific bacteria present on the eggs [22]. Similarly, Lam et al. (2009) found that the bacteria present on the eggs of *M. domestica* act as a nutrient reserve [30], a phenomenon which may explain the reduced final biomass observed in this study.

Growth for disinfected eggs did not seem to be influenced in this current study. While significant differences across days were observed within treatments, the effect size for disinfected eggs and control were similar (GES: 0.768 and 0.739, respectively), suggesting that the major factor for variance was time for both treatments.

This is not in accordance with a different study by Lam et al. (2009) which uncovered that the presence of bacterial colonies on eggs was associated with the suppression of fungal growth in their substrate and showed that the removal of said bacteria increased fungal growth, which, inadvertently, affected *M. domestica* adult fly emergence negatively [31]. While this current study did not investigate emergence, the presence of fungicidal bacteria could have affected the growth of the larvae. Although these findings are not supported in this study, they might still have contributed to the observed difference in final biomass. As mentioned, this study found no significant difference in survival rate between the disinfected- and the non-disinfected eggs. However, this is not in accordance with the study by Lam et al. (2009), who observed the decreased survival of sterilised eggs [31]. This could be explained by the fact that insects contain egg-associated microbiota which assists with the suppression of fungi in the substrate and protect against pathogens [10,31]. Therefore, it could be speculated that the removal of microbiota from the egg-surface would reduce the survival rate. However, this study does not find a correlation between survival and egg-associated microbiota.

### 4.2. The Effect of Autoclaved Substrate

It was expected that the autoclaved substrate would lead to decreased growth, final biomass, and survival rate. This current study found a significant difference between autoclaved and non-autoclaved substrates across different days with respect to growth, as well as reduced final biomass and survival rate.

Similarly, Greenwood et al. (2021) observed that the microorganisms in the substrate significantly affected the final biomass and the fat and protein content of *H. illucens* larvae [32]. This is in accordance with previous studies, which showed that microorganisms from the substrate are connected to gut microbial composition, as 66% of all microorganisms in *H. illucens* larvae’s gut originated from the substrate [33]. These concepts give insights into why larvae were affected negatively when microorganisms were removed from the substrate in this current study. Therefore, the differences in growth, final biomass, and survival rate observed were not unexpected, considering the removal of essential microorganisms.

The decrease in biomass and survival could also be due to decreased availability of nutrients as a result of the Maillard reaction, which might occur during autoclaving. This would lead to a decrease in the concentration of amino acids, but also to a reduced digestibility of amino acids [34]. Another study by Hefnawy (2011) showed that autoclaving lentils (*Lens culinaris* Medikus) led to a decrease in different nutrients like sucrose and raffinose [35]. Therefore, it is possible that the autoclaved substrate in the current study lacked important nutrients that may affect both microorganisms and the larvae. These principles help explain the observed reduction in final biomass and decreased survival rate. Moreover, they might explain the impaired growth, which is supported by the lower effect size compared to the control treatment (GES: 0.36 and 0.739, respectively). This suggests that factors other than time are influencing the variance observed throughout the growth process. Somroo et al. (2019) also determined the positive effects of microorganisms on *H. illucens* larvae. With soybean curd residues as a growth substrate for the larvae and the presence of *Lactobacillus buchneri*, parameters such as final biomass and fat and protein content were found to increase compared to those associated with the larvae grown in the absence of *L. buchneri* [36].

### 4.3. The Effect of Disinfected Eggs and Autoclaved Substrate

It was expected that disinfection of eggs and autoclaving substrate would lead to decreased growth, final biomass, and survival rate. This current study found a significant difference in terms of larval growth between the disinfected eggs in autoclaved substrate with a low effect size and the control (GES: 0.07 and 0.739, respectively), indicating that factors other than time influenced growth. A significant difference in final biomass and survival rate was found when comparing the experimental treatments with the control. The reasons for these findings might be the accumulation of all the previously listed effects of reduced nutrient digestibility and loss of beneficial microbiota.

### 4.4. Perspectives

It is not certain that the disinfected eggs and the autoclaved substrate were uncontaminated. It is important to note that, despite autoclaving, the substrate still showed the presence of some microbial colonies, as observed in the agar plate analysis (see Appendix B). Autoclaving may not achieve complete sterilisation. Possible reasons for this could include the survival of heat-resistant spores or contamination post-autoclaving during handling or exposure to the air. This could have affected the results specifically through interactions with microbes and nutrient availability. Agar plates (see Appendix B) revealed that the autoclaved substrate had fewer colonies compared to the non-autoclaved counterpart. However, there were no visible colonies in the disinfected eggs sample.

Furthermore, this study does not examine species present on the eggs and in the substrate; therefore, performing a 16S ribosomal RNA analysis could be beneficial to evaluate both the number of bacteria and the species present in the substrate and on the eggs. Specifically, which species are present and their relative contribution to biomass, survival rate, and growth could be of great interest for commercial production. The results show the importance of allowing insects to vertically transmit and horizontally inherit microorganisms, both of which result in increased growth and survival rate which are beneficial to maximise insect production.

Future studies are necessary to understand the full scope of the specific bacteria which are relevant for the aforementioned parameters. Moreover, it could be interesting to investigate which physical conditions (pH, temperature, humidity, etc.) together with the host and host microbiome are optimal for enhancing biomass and survival rate for production. Also, to investigate which substrates are optimal to increase the growth, final biomass, and survival rate to maximize the outcome. Alternatively, it could be valuable to conduct a similar experiment on wild caught flies and determine if these findings translate to such a population.

## 5. Conclusions

In conclusion, disinfection of eggs and autoclaving of the substrate both had a significant effect on the final biomass of the larvae of *Musca domestica*. The survival rate decreased significantly in the treatments with both disinfected eggs and autoclaved substrate and in the treatments with only the autoclaved substrate, but not in the treatments with only disinfected eggs. The growth of larval biomass was also influenced by treatment, with 76.8% and 73.9% of the variance explained by the time factor for the treatment with only disinfected eggs and the control, respectively. For autoclaved substrate and both disinfected and autoclaved eggs and substrate, 36% and 7% of the variance, respectively, was explained by time, indicating that other factors were affecting growth. Further research on more commercially relevant substrates would be crucial to uncover the full effect of microorganisms in insect production.

## Figures and Tables

**Figure 1 insects-15-00764-f001:**
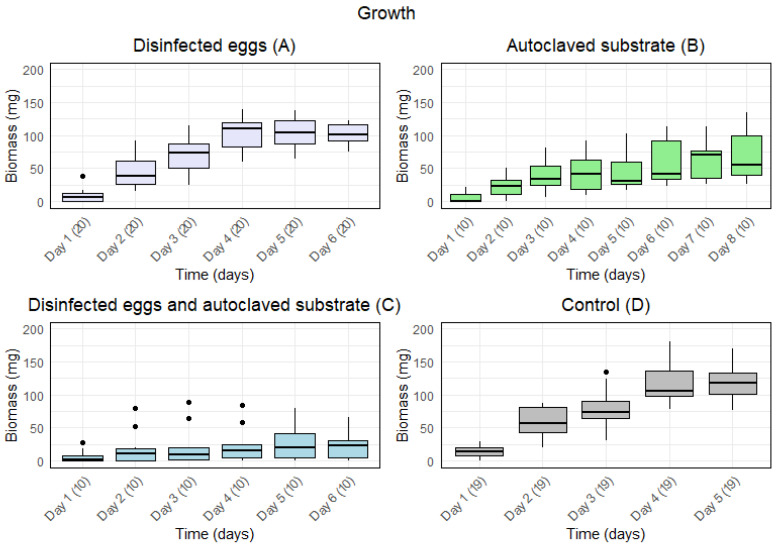
Boxplots with the medians (25–75% quantiles) of the biomass of larvae from day 1 and until the first five larvae were present in one sample for treatment: (**A**) disinfected eggs, (**B**) autoclaved substrate, (**C**) disinfected eggs and autoclaved substrate, and (**D**) control. The *x*-axis shows time (days) with sample size in parentheses, and the *y*-axis shows the biomass (mg).

**Figure 2 insects-15-00764-f002:**
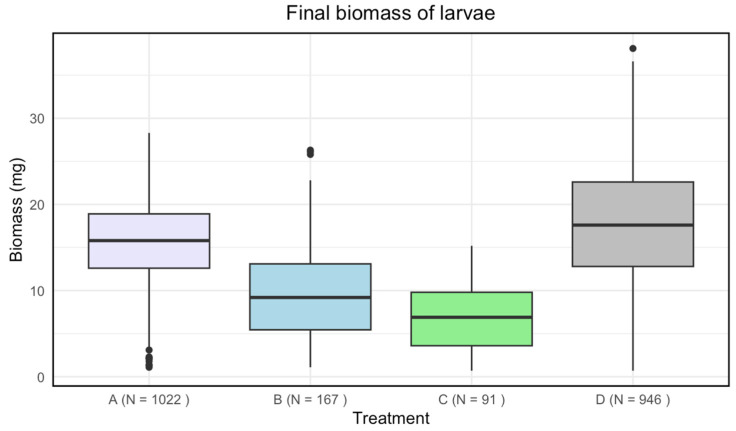
Boxplots with the medians (25–75% quantiles) of the final biomass of larvae for treatment A (disinfected eggs), B (autoclaved substrate), C (disinfected eggs and autoclaved substrate), and D (control). The *x*-axis shows treatments and sample size (N), and the *y*-axis shows the biomass (mg).

**Figure 3 insects-15-00764-f003:**
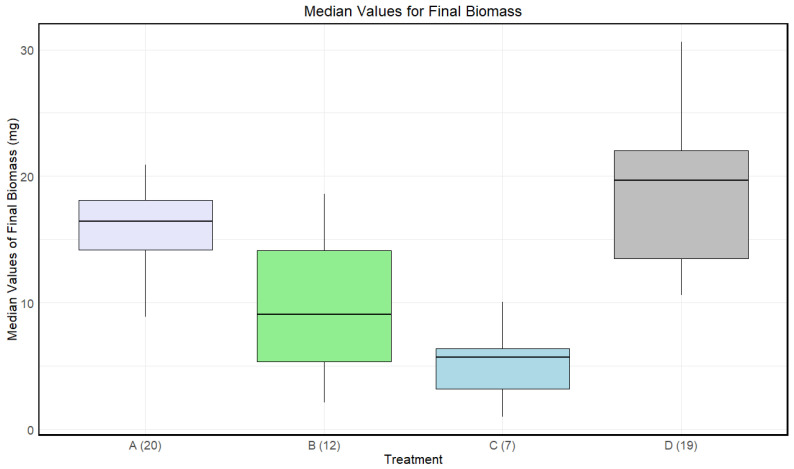
Boxplots with the medians (25–75% quantiles) of the medians of each sample of larvae for treatment A (disinfected eggs), B (autoclaved substrate), C (disinfected eggs and autoclaved substrate), and D (control). The *x*-axis shows treatments and sample size in parentheses, and the *y*-axis shows the survival rate (%).

**Figure 4 insects-15-00764-f004:**
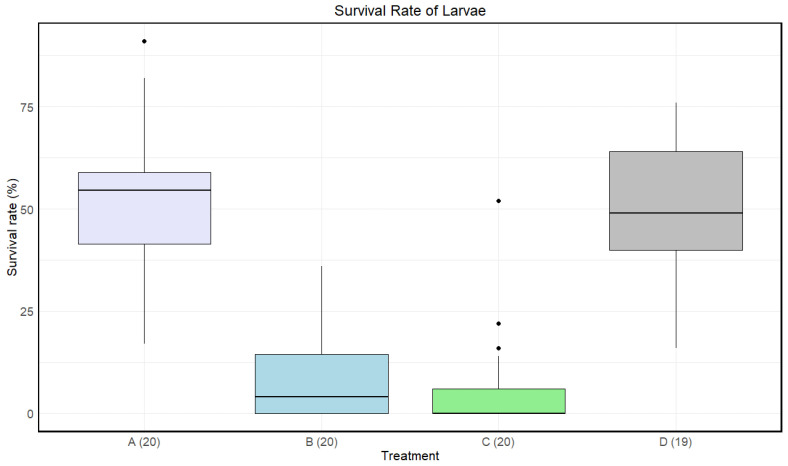
Boxplots with the medians (25–75% quantiles) of the survival rate of larvae for treatment A (disinfected eggs), B (autoclaved substrate), C (disinfected eggs and autoclaved substrate), and D (control). The *x*-axis shows treatments and sample size in parentheses, and the *y*-axis shows the survival rate (%).

**Table 1 insects-15-00764-t001:** Repeated measures ANOVA including *p*-value, F-value, and generalised eta squared (GES) for treatment A (disinfected eggs), B (autoclaved substrate), C (disinfected eggs and autoclaved substrate), and D (control). Significant *p*-values are indicated with an asterisk (*) when *p* < 0.05.

Treatment	*p*-Value	F-Value	GES
Disinfected eggs (A)	6.44·10^−28^ *	159.253	0.768
Autoclaved substrate (B)	1.2·10^−5^ *	14.481	0.36
Disinfected eggs and autoclaved substrate (C)	0.028 *	4.375	0.07
Control (D)	3.54·10^−37^ *	3186.309	0.739

**Table 2 insects-15-00764-t002:** A Kruskal–Wallis indicated a significant effect of treatments when comparing all data points in each sample. A Dunn’s test was then performed between pairs of treatments, A (disinfected eggs), B (autoclaved substrate), C (disinfected eggs and autoclaved substrate), and D (control). Significant adjusted *p*-values are indicated with an asterisk (*) when *p* < 0.05.

Comparison	Unadjusted	Adjusted *p*-Value
A–B	$1.18\cdot 10^{-23}$ *	$7.05\cdot 10^{-23}$ *
A–C	$2.26\cdot 10^{-31}$ *	$1.36\cdot 10^{-30}$ *
B–C	$7.79\cdot 10^{-4}$ *	$4.68\cdot 10^{-3}$ *
A–D	$1.17\cdot 10^{-11}$ *	$7.03\cdot 10^{-11}$ *
B–D	$3.21\cdot 10^{-42}$ *	$1.93\cdot 10^{-41}$ *
C–D	$5.04\cdot 10^{-47}$ *	$3.03\cdot 10^{-46}$ *

**Table 3 insects-15-00764-t003:** A Kruskal–Wallis indicated a significant effect of treatments, when comparing the median of each sample. A Dunn’s test was then performed between pairs of treatments, A (disinfected eggs), B (autoclaved substrate), C (disinfected eggs and autoclaved substrate), and D (control). Significant *p*-values are indicated with an asterisk (*) when *p* < 0.05.

Comparison	Unadjusted *p*-Value	Adjusted *p*-Value
A–B	$1.1\cdot 10^{-2}$ *	$6.7\cdot 10^{-2}$
A–C	$4.1\cdot 10^{-4}$ *	$2.5\cdot 10^{-3}$ *
B–C	$1.89\cdot 10^{-1}$	1.00
A–D	$1.72\cdot 10^{-1}$	1.00
B–D	$2.2\cdot 10^{-4}$ *	$1.3\cdot 10^{-3}$ *
C–D	$6.85\cdot 10^{-6}\;$ *	$4.1\cdot 10^{-4}$ *

**Table 4 insects-15-00764-t004:** A Kruskal–Wallis indicated a significant effect of all treatments. A Dunn’s test was then performed between pairs of treatments, A (disinfected eggs), B (autoclaved substrate), C (disinfected eggs and autoclaved substrate), and D (control). Significant adjusted *p*-values are indicated with an asterisk (*) when *p* < 0.05 for both tests.

Comparison	Unadjusted *p*-Value	Adjusted *p*-Value
A–B	$3.47\cdot 10^{-7}$ *	$2.08\cdot 10^{-6}$ *
A–C	$1.10\cdot 10^{-8}$ *	$6.58\cdot 10^{-8}$ *
B–C	$2.63\cdot 10^{-1}$ *	1
A–D	$4.52\cdot 10^{-1}$ *	1
B–D	$3.80\cdot 10^{-7}$ *	$5.28\cdot 10^{-6}$ *
C–D	$3.26\cdot 10^{-8}$ *	$1.96\cdot 10^{-7}$ *

## Data Availability

The data presented in this study are available upon request from the corresponding author.

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
