# Peer review of "The Effects of Egg- and Substrate-Associated Microbiota on the Larval Performance of the Housefly, Musca domestica"

_insects, 2024, doi:10.3390/insects15100764_

Round 1
Reviewer 1 Report
Comments and Suggestions for Authors
Dear authors,
the topic of your research is very interesting. Unfortunately I am regret to inform that your article is not of a standard acceptable for publication its present form and major changes are required. I strongly recommend to resubmit your manuscript once the necessary changes. I hope my suggestions can be helpful in the publication of the manuscript.
The result section should strongly be improved. Figures and tables are currently difficult to understand. I suggest replacing the title of the graph for a more immediate interpretation of the results. For example, replace "Development of biomass (A)" with "Disinfected eggs). Also in the tables it would be appropriate to indicate the treatments in full and not in letters.
- The graphs included in the manuscript and the statistical analysis reported seem to report different information. Could it be that in your test minimal differences were assessed as significant perhaps because of the high number of repetitions? Including effect size could help to clarify this point.
- The length of the introduction is excessive. Form the beginning up to line 99, the introduction can be greatly reduced and be more relevant to your research.
- Scientific names must be given in full (author, order and family must be indicated) the first time they are mentioned in the abstract and in the manuscript.
-The hatching rates in the control (0% w/w) is below 50% (appendix A). Could you explain this data?
- line 250: specify if larvae have been weighed prior freezing.
Author Response
AD1) The result section should strongly be improved. Figures and tables are currently difficult to understand. I suggest replacing the title of the graph for a more immediate interpretation of the results. For example, replace "Development of biomass (A)" with "Disinfected eggs). Also in the tables it would be appropriate to indicate the treatments in full and not in letters
RE: We agree, on the fact the figures and tables could be more clear and have changed them according to the Reviewer’s comment.
AD2) The graphs included in the manuscript and the statistical analysis reported seem to report different information. Could it be that in your test minimal differences were assessed as significant perhaps because of the high number of repetitions? Including effect size could help to clarify this point.
RE: Thank you for your insightful comments regarding the potential inflation of significant differences due to the large sample size. We have added a new graph and table to further clarify our point. See figure 4 and table 4.
AD3) The length of the introduction is excessive. Form the beginning up to line 99, the introduction can be greatly reduced and be more relevant to your research.
RE: We agree the introduction contains information that doesn’t fit within the interest of this study, and we have now removed sections which are not as relevant.
AD4) Scientific names must be given in full (author, order and family must be indicated) the first time they are mentioned in the abstract and in the manuscript.
RE: This has been added.
AD5)The hatching rates in the control (0% w/w) is below 50% (appendix A). Could you explain this data?
RE: The hatching rate of the control is 50%. This simply means that 50% of the total eggs collected, so 50 eggs, survived to the larval stage. Therefore approx. 23 eggs made it to larval stage. This is within the range of what can be expected for Musca domestica.
AD6) line 250: specify if larvae have been weighed prior freezing.
RE: Yes, this has been specified, changed to include that they were weighed after being frozen. See line 218
Reviewer 2 Report
Comments and Suggestions for Authors
My opinion is that these are confirmatory results in relation to practices which, to my knowledge, are not used. Disinfecting eggs and/or substrate is of no economic, zootechnical or sanitary interest. The reviewer encourages the authors to better decipher natural microbiota under real-life conditions, in order to identify flora of interest for one or more of the desired effects (biomass, survival...).
Author Response
AD1) My opinion is that these are confirmatory results in relation to practices which, to my knowledge, are not used. Disinfecting eggs and/or substrate is of no economic, zootechnical or sanitary interest. The reviewer encourages the authors to better decipher natural microbiota under real-life conditions, in order to identify flora of interest for one or more of the desired effects (biomass, survival...).
RE: Thank you for this comment. We agree that the findings of the this study have implications for studying insect-microbiota interactions in general, but also from an applied point of view. We also agree that studies in “real” life is needed to make firm conclusions regarding the use of disinfecting eggs and/or substrate to improve economic, zootechnical or sanitary focus. Moreover, we wanted to uncover the magnitude of microbiota in a commercial substrate and what the detriments could be if the microbiota is reduced. The plan for future projects is also to uncover specific bacteria and their contributions to the mentioned effects.
We have changed the perspectives section to specify .We included the perspective of conducting a similar experiment on wild caught flies. See line 431-432
Reviewer 3 Report
Comments and Suggestions for Authors
The manuscript “The effects of the egg- and substrate-associated microbiota on the larval performance of the housefly, Musca domestica” can represent a contribution of practical knowledge in a phase of housefly breeding; as the role of bacteria is already known. The experimentation carried out is fundamentally simple but, in some parts, it is written in an unnecessarily verbose repetitive way. A better flow of information would facilitate the reader. The materials and methods are sufficiently detailed, but only if the first experiment is rewritten as a preliminary test (it has many experimental gaps). The presence of colonies in the autoclaved substrate raises doubts about the work performed in sterility, and should be considered in discussions even if it probably did not affect the results.
Below are the specific comments:
Simple summary: please, check the format;
Key words: I suggest replacing keywords already present in the title (Musca domestica, larval performance);
INTRODUCTION
Line 81: the citation [8] does not consider the housefly. I suggest rewriting the sentence or using a more appropriate citation;
Line 114: citations are usually listed in ascending order (check throughout the text);
Line 151-153: this sentence is M&M. I suggest removing it;
MATERIALS AND METHODS
The reduction of the chapters and their reorganization would improve readability and highlight the main experiment (the current second Experiment). It is my opinion that Experiment 1 is a preliminary test to establish the dose of sodium hypochlorite to be used in the subsequent experiment (in fact, the experimental design, replications, statistical analysis, etc. is not clear).
Line 161: please, correct to “23 ± 1 °C”;
Line 162: please, replace “flies” with “larvae”;
Line 165: if it is a commercial bugdorm, add the supplier and its country, otherwise add the dimensions;
Line 165-166: please rewrite the sentence more clearly;
Line 174: please, reconsider it as a preliminary test;
Line 179: I suggest reporting the concentrations in the text and eliminating the table 1;
Line 180: I suggest describing section 2.2.1 here, and not in a separate later chapter;
Line 182: I suggest describing section 2.2.2 here, and not in a separate later chapter;
Line 212: please, the concentration of chlorine bleach used must be added;
Line 219: please, add space after "120", and add the time (for completeness);
Line 226: please add the country to the company;
Line 241: this chapter can be included in the next one;
RESULTS
Line 273: please, correct figure 4.1 with figure 1
Line 300: please, correct figure 4.2 with figure 2
Line 317: please, correct figure 4.3 with figure 3
DISCUSSION
Line 394: in this part, the authors should consider and justify the presence of colonies in the autoclaved substrate (Table B.1);
Line 410-411: generic and therefore superfluous sentence. Please, delete them.
421-446: the perspectives should be more concise and focused on the application aspects of the results in industrial fly farming.
Appendix A
Table A.1: please, check the concentration 0.001% (it is probably 0.01%);
Figure A1: the plate covers show confusion. If possible, I suggest replacing the photos.
Author Response
AD1) Simple summary: please, check the format;
RE: This has been corrected
AD2) Key words: I suggest replacing keywords already present in the title (Musca domestica, larval performance);
RE: Thank you for this suggestion. This has been changed to include “microbial effects, survival rate, biomass, development.”
AD3) Line 81: the citation [8] does not consider the housefly. I suggest rewriting the sentence or using a more appropriate citation;
RE: The sentence has been deleted.
AD4) Line 114: citations are usually listed in ascending order (check throughout the text);
RE: Thank you. This mistake has been corrected throughout the text.
AD5) Line 151-153: this sentence is M&M. I suggest removing it;
RE: We agree this belongs more to M&M and it has therefore been moved.
AD6) MATERIALS AND METHODS
The reduction of the chapters and their reorganization would improve readability and highlight the main experiment (the current second Experiment). It is my opinion that Experiment 1 is a preliminary test to establish the dose of sodium hypochlorite to be used in the subsequent experiment (in fact, the experimental design, replications, statistical analysis, etc. is not clear).
RE: Thank you for this suggestion. It is correct that Experiment 1 is a preliminary to test to examine dosage. We have therefore changed the title of the experiment. Furthermore, this section has been rearranged and should be clearer in experimental design and replications. Moreover, the result of the test is covered in Appendix A.
AD7) Line 161: please, correct to “23 ± 1 °C”
RE: This has been changed accordingly.
AD8) Line 162: please, replace “flies” with “larvae”
RE: Yes, this has been corrected.
AD9) Line 165: if it is a commercial bugdorm, add the supplier and its country, otherwise add the dimensions;
RE: This has been changed, and now includes supplier and its country at line 139-140
AD10) Line 165-166: please rewrite the sentence more clearly;
RE: This has been changed to include it was adult flies who were fed water and the mixture ad libitum. See line 140-142
AD11) Line 174: please, reconsider it as a preliminary test;
RE: See earlier comment regarding your suggestion on experiment 1.
AD12) Line 179: I suggest reporting the concentrations in the text and eliminating the table 1;
RE: We eliminated table 1 and reported the concentrations in the text as per your suggestion. See line 153
AD13) Line 180: I suggest describing section 2.2.1 here, and not in a separate later chapter;
RE: We agree with this suggestion and has changed it accordingly
AD14) Line 182: I suggest describing section 2.2.2 here, and not in a separate later chapter;
RE: We agree and have changed accordingly.
AD15) Line 212: please, the concentration of chlorine bleach used must be added;
RE: It has been incorporated at line 154-156
AD16) Line 219: please, add space after "120", and add the time (for completeness);
RE: This has been added.
AD17) Line 226: please add the country to the company;
RE: This has been changed to include the countries. See line 174-176
AD18) Line 241: this chapter can be included in the next one;
RE: We agree and have changed accordingly.
AD19) RESULTS
Line 273: please, correct figure 4.1 with figure 1
RE: This has been changed.
AD20) Line 300: please, correct figure 4.2 with figure 2
RE: This has been changed.
AD21) Line 317: please, correct figure 4.3 with figure 3
RE: This has been changed.
AD22) DISCUSSION
Line 394: in this part, the authors should consider and justify the presence of colonies in the autoclaved substrate (Table B.1);
RE: this have been addressed at line 409-417. It now includes the section “It is important to note that, despite autoclaving, the substrate still showed the presence of some microbial colonies, as observed in the agar plate analysis (see Appendix B.1). Autoclaving may not achieve complete sterilization. Possible reasons for this could include the survival of heat-resistant spores or contamination post-autoclaving during handling or exposure to air. This could have affected the results specifically through interactions with microbes and nutrient availability”
AS23) Line 410-411: generic and therefore superfluous sentence. Please, delete them.
RE: We agree, and the sentence has been deleted
AD24) 421-446: the perspectives should be more concise and focused on the application aspects of the results in industrial fly farming.
RE: This section has been changed and should now put greater emphasis on our results in relation to the industry, but also limitations (see earlier response).
It now reads, “Furthermore, this study does not examine species present on the eggs and in the substrate, therefore, performing a 16S ribosomal RNA analysis could be beneficial in evaluating both the number of bacteria and the species present in the substrate and on the eggs, moreover, uncovering which species are present and their relative contributions to biomass, survival rate and development could be of great interest for commercial production. The results show the importance of allowing insects to vertically transmit and horizontally inherit microorganisms which results in increased growth and survival rate which are beneficial in maximising insect production. ”.
AD25) Appendix A
Table A.1: please, check the concentration 0.001% (it is probably 0.01%);
RE: Yes, this has been corrected.
AD26) Figure A1: the plate covers show confusion. If possible, I suggest replacing the photos.
RE: We do agree that the covers can be confusing. Unfortunately, we did not take any more photos, as we were unsure if this was completely necessary for the preliminary test, but did decide they were worth including regardless.
Reviewer 4 Report
Comments and Suggestions for Authors
Dear authors,
the manuscript is well written and clear and i do believe that the perspectives section is really relevant. I might have to say that even if the research lack of experiments related to microbial identification overall the content is well presented. I do suggest to resubmit the paper after analyzing the data with other statistical test.
Kind regards
Author Response
AD1) Dear authors,
the manuscript is well written and clear and i do believe that the perspectives section is really relevant. I might have to say that even if the research lack of experiments related to microbial identification overall the content is well presented. I do suggest to resubmit the paper after analyzing the data with other statistical test.
Kind regards
RE: We think that the repeated measures ANOVA, Wilcoxon, and Dunn’s test statistical tests, used in the manuscript are the right ones and we have obtained very clear results without problems of statistical power, therefore unless that the Reviewer have something specific in mind we would like to not change our statistical pipeline.
Reviewer 5 Report
Comments and Suggestions for Authors
Housefly Musca domestica is a dipteran insect which larvae considered as a promising alternative to recycle organic food waste and by-products. The authors of this manuscript investigated the effect of egg- and subtsrate-associated microorganisms on the housefly larvae’s performance. The overall manuscript was well-written. I only have very minor comments or suggestions below.
Abstract
Lines 31-34
Why does this have anything to do with microorganisms in housefly? There is a missing sentence here before “Microorganisms …” In other words, the first sentence does not lead to the second sentence.
Entire document
Please check editorial requirements such as proper use of comma, grammar etc. For example, in Figure B.1. (line 493):
3 refer to …à should be 3 refers to …
Entire References
Need consistency on format, especially on DOI and website link à does it need to be underlined or not etc. Need to check with the publisher guidelines.
Author Response
AD1) Lines 31-34
Why does this have anything to do with microorganisms in housefly? There is a missing sentence here before “Microorganisms …” In other words, the first sentence does not lead to the second sentence.
RE: We agree that there could be a better transition to our main focus. This has now been changed at line 35. It now reads: “ Insects are an interesting and more sustainable alternative to conventional animal source proteins as they can convert waste- and by-products from the agricultural industry to biomass for commercial feed for livestock and potentially as a food source for human consumption. Moreover, microorganisms have been shown to play a pivotal role in the development of insects, breakdown of complex growth substrates, and are therefore closely tied to insect production”
AD2) Please check editorial requirements such as proper use of comma, grammar etc. For example, in Figure B.1. (line 493):
3 refer to …à should be 3 refers to …
RE: Thank you for this. The article has been thoroughly read through with comma and general grammar in focus.
AD2) Entire References
Need consistency on format, especially on DOI and website link à does it need to be underlined or not etc. Need to check with the publisher guidelines.
RE: Yes, there were definitely some errors. These have now been corrected.
Round 2
Reviewer 2 Report
Comments and Suggestions for Authors
Ok for publication
Reviewer 4 Report
Comments and Suggestions for Authors
Dear authors,
thank you for your reply. You are right the statistic is well run.
Kind regards